# Miracle Berry as a Potential Supplement in the Control of Metabolic Risk Factors in Cancer

**DOI:** 10.3390/antiox9121282

**Published:** 2020-12-15

**Authors:** Marta Gómez de Cedrón, Sonia Wagner, Marina Reguero, Adrián Menéndez-Rey, Ana Ramírez de Molina

**Affiliations:** 1Molecular Oncology Group, Precision Nutrition and Health, IMDEA Food Institute, CEI UAM + CSIC, Ctra. de Cantoblanco 8, 28049 Madrid, Spain; sonia.wagner@imdea.org (S.W.); marina.reguero@imdea.org (M.R.); 2Medicinal Gardens SL, Marqués de Urquijo 47, 28008 Madrid, Spain; adrian.menendez.rey@alumnos.upm.es; 3NATAC BIOTECH, Electronica 7, Alcorcón, 28923 Madrid, Spain; 4Biomedical Technology Center, Polytechnic University of Madrid, 28223 Pozuelo de Alarcón, Spain

**Keywords:** oxidative stress, chronic diseases, cancer, miracle berry, miraculin, gut–brain axis

## Abstract

The increased incidence of chronic diseases related to altered metabolism has become a social and medical concern worldwide. Cancer is a chronic and multifactorial disease for which, together with genetic factors, environmental factors are crucial. According to the World Health Organization (WHO), up to one third of cancer-related deaths could be prevented by modifying risk factors associated with lifestyle, including diet and exercise. Obesity increases the risk of cancer due to the promotion of low-grade chronic inflammation and systemic metabolic oxidative stress. The effective control of metabolic parameters, for example, controlling glucose, lipid levels, and blood pressure, and maintaining a low grade of chronic inflammation and oxidative stress might represent a specific and mechanistic approach against cancer initiation and progression. Miracle berry (MB) (*Synsepalum dulcificum)* is an indigenous fruit whose small, ellipsoid, and bright red berries have been described to transform a sour taste into a sweet one. MB is rich in terpenoids, phenolic compounds, and flavonoids, which are responsible for their described antioxidant activities. Moreover, MB has been reported to ameliorate insulin resistance and inhibit cancer cell proliferation and malignant transformation in vitro. Herein, we briefly summarize the current knowledge of MB to provide a scientific basis for its potential use as a supplement in the management of chronic diseases related to altered metabolism, including obesity and insulin resistance, which are well-known risk factors in cancer. First, we introduce cancer as a metabolic disease, highlighting the impact of systemic metabolic alterations, such as obesity and insulin resistance, in cancer initiation and progression. Next, as oxidative stress is closely associated with metabolic stress, we also evaluate the effect of phytochemicals in managing oxidative stress and its relationship with cancer. Finally, we summarize the main biological activities described for MB-derived extracts with a special focus on the ability of miraculin to transform a sour taste into a sweet one through its interaction with the sweet taste receptors. The identification of sweet taste receptors at the gastrointestinal level, with effects on the secretion of enterohormones, may provide an additional tool for managing chronic diseases, including cancer.

## 1. Cancer as a Metabolic Disease

Cancer is the second leading cause of death worldwide (WHO, 2018). In addition to genetic alterations, additional factors related to lifestyle, such as diet (high intake of saturated fatty acids, red meat consumption, and high glucose intake), a sedentary lifestyle, and obesity are considered to be preventable risk factors [1]. During the last few years, the global prevalence of overweight and obesity has increased alarmingly [2]. Obesity represents a risk factor for many chronic diseases, such as cardiovascular disease (CDV), dyslipidemia, metabolic syndrome (MetS), type 2 diabetes mellitus (T2DM), non-alcoholic fatty liver disease (NAFLD), and cancer [3]. Over the course of obesity, ectopic fat deposition (known as visceral adiposity) leads to a low grade of chronic inflammation, insulin resistance, and an altered adipokine profile, which can shape a pro-tumoral microenvironment supporting malignant transformation and progression [4].

Although the role of obesity in cancer etiopathogenesis is not fully elucidated, the World Cancer Research Fund and the American Institute for Cancer Research (WCRF/AICR) have found strong evidence for the association between obesity and the risk of several cancers, including esophageal cancer, colorectal cancer, liver cancer, pancreatic cancer, postmenopausal breast cancer, and kidney cancer [5]. Interestingly, epidemiological data support the idea that obesity may be a protective factor for certain types of cancer with regard to incidence and mortality, such as premenopausal breast cancer (BC), non-small cell lung cancer (NSCLC), head and neck cancers, renal cell cancer, and metastatic colorectal cancer (CRC) [6]. Potential explanations for the obesity paradox in cancer patients may be related to inadequate anthropometric measurements of obesity (body mass index (BMI) versus visceral adipose tissue) or cofounding factor adjustments (sex, diet, and exercise). In this regard, BMI cannot differentiate between adipose and lean mass or visceral adipose tissue, resulting in contradictory effects between obesity and cancer [4].

The main metabolic pathways linking obesity to cancer include oxidative stress, a low-grade of chronic inflammation, hyperinsulinemia, alterations in the adipocytokine profile, and intestinal dysbiosis [7].

### 1.1. Obesity Is Associated with a Low Grade of Chronic Inflammation and Cancer Risk

Obesity-associated low-grade chronic inflammation is an established mediator for cancer initiation and progression [8]. Adipose tissue is an active endocrine organ, secreting adipokines to control the whole body’s energy balance. Excessive ectopic fat distribution is associated with increased levels of leptin, which stimulates the production of proinflammatory mediators such as IL-1, IL-6, IL-8, TNF-α, and cyclooxygenase 2 (COX2). On the contrary, levels of the anti-inflammatory adiponectin are reduced [9].

Obesity has a causal relation with insulin resistance (IR), which is also closely associated with a low grade of chronic inflammation. A recent study by Lee et al. attempted to elucidate the effect of systemic inflammation on all-cause and cancer-related mortality. It was found that individuals with elevated HOMA-IR (homeostatic model assessment for insulin resistance) or high sensitivity C-reactive protein (hsCRP) had increased mortality rates, with the effect of systemic inflammation in cancer-related mortality being more pronounced [10]. The increased circulating levels of glucose and free fatty acids during obesity contribute to augmenting the levels of ROS, which may induce mitochondrial and DNA damage [11].

The excess adipose tissue in cancer shapes the local tumor microenvironment, promoting tumor dissemination, angiogenesis, and the inhibition of antitumor immune responses [12].

One additional factor to be considered for the association between cancer and obesity is the altered intestinal microbiome (dysbiosis), which has been observed in obese individuals as compared with lean individuals. The host organism is a complex system of innate immunity and intestinal epithelial cells that control invasion through a robust gut barrier. Alterations in the composition and function of the gut microbiota disrupt the equilibrium with the host and contribute to the promotion of intestinal inflammation, which is closely associated with inflammatory bowel disease (IBD), diabetes, obesity, and cardiovascular diseases (CVDs). The diet and metabolic statuses of individuals are key factors affecting the diversity and composition of the microbiome [13,14]. In this way, effective management of the nutritional and metabolic status at the systemic level contribute to promoting a healthy gut microbiome, thereby diminishing the risk of intestinal inflammatory diseases such as ulcerative colitis, inflammatory bowel disease, and cancer [15,16,17,18]. The microbiota have been demonstrated to influence body weight control by affecting the extraction and absorption of calories, influencing the secretion of anorexigenic hormones (GLP-1, PYY, and leptin), and controlling the intestinal gut barrier [19]. Firmicutes and Bacteroidetes are dominant in the gut microbiota, corresponding to 90% of bacteria. Bacteroidetes have a positive correlation with a reduction in body fat, whereas the relationship between Firmicutes and obesity is associated with greater energy harvest [20].

Importantly, the gut microbiota can be easily affected, with diet among the main factors influencing the gut microbiota’s composition. Moreover, the metabolites produced by the microbiota, such as short-chain fatty acids, can exert different effects on the host’s metabolism. Short-chain fatty acids influence the metabolism of lipids, glucose, and cholesterol, affecting adipose tissue’s fat storage and the immune system’s function [21,22].

Although gut microbiota are undeniably related to obesity, and weight loss interventions certainly lead to changes in the microbiota, our understanding of the role of these bacteria in obesity remains insufficient for designing specific and personalized nutritional interventions [23] (Figure 1).

### 1.2. Dual Role of Phytochemicals Modulating Oxidative Stress in Cancer

Several epidemiological observations have shown an inverse relationship between the consumption of plant-based foods rich in phytochemicals and the incidence of cancer. Phytochemicals, due to their antioxidant, activity play a key role in cancer chemoprevention by suppressing the DNA damage induced by excessive oxidative stress.

Oxidative stress can promote tumor initiation and growth by introducing mutations in oncogenes and tumor suppressors. Therefore, on the one hand, reducing oxidative stress may protect normal cells from carcinogenic transformations. On the other hand, cancer cells are characterized by increased intrinsic oxidative stress. Thus, cancer cells rely on antioxidants for their survival.

Reactive oxygen species (ROS) are highly reactive metabolic by-products that cause both deleterious and beneficial effects. Although cellular ROS are mediators of the cell signaling pathways required for normal physiological functions such as differentiation and development [24], the overproduction of ROS may be detrimental, as ROS can disrupt cell integrity and promote cell damage [25]. In this way, on the one hand, excessive ROS may contribute to the initiation of the carcinogenic process or the appearance of resistance to therapeutic treatments. On the other hand, excessive accumulation of ROS might induce cancer cell death. Many studies have shown that cancer cells have increased levels of ROS as compared with normal cells due to their high metabolic rates or mitochondrial dysfunctions. In this way, agents augmenting oxidative stress in cancer may result in the promotion of specific cancer cell death [26]. In this scenario, increasing oxidative stress using agents with the capacity to promote ROS [27] and the abrogation of key antioxidant systems [28,29] may make cancer cells vulnerable to cell death.

Phytochemicals have been shown to exhibit both antioxidant and pro-oxidant effects in cancer depending on different factors, including the implicated oncogenic and tumor suppressors, the stage of the carcinogenic process, the heterogeneity of tumors, and the associated tumor microenvironment. In addition, the antioxidant and pro-oxidant properties of phytochemicals may also depend on their concentration [26,30].

In this regard, by acting as pro-oxidant agents, several natural dietary bioactive compounds, including polyphenols, flavonoids, and stilbenes may promote, in a specific manner, cancer cell death [26]. While most dietary bioactive compounds possess antioxidant capacities at low doses, high doses induce pro-oxidant activity that leads to cancer cell death. Although the specific mechanisms in the process are not fully elucidated, pro-oxidant agents mainly influence mitochondrial functions by altering mitochondrial enzymes and oxidative phosphorylation [31,32].

## 2. Miracle Berry

*Synsepalum dulcificum* (*Richardella dulcifica*) is an indigenous fruit from tropical West Africa. The edible pulp of the small, ellipsoid, and bright red berry of this plant converts a sour taste into a sweet taste; for this reason, the fruit is also known as the “miracle berry” (MB) [33].

MB is rich in terpenoids. The fruit’s pulp contains phenolic compounds (15.8%) and flavonoids (11.9%), while its skin contains higher amounts of phenolic compounds (36.7%) and flavonoids (51.9%). There is a direct correlation between phenolic and flavonoid content and the antioxidant activity; therefore, the skin has higher antioxidant activity (22.6%) as compared with that of the pulp (18.9%) [34,35].

MB, which is mainly associated with its antioxidant activity, has been demonstrated to reduce malignant cell proliferation in vitro [35,36]. Moreover, miraculin present in the fruit has the ability to transform a sour taste into a sweet one [37,38,39]. The taste-modifying effect of MB is effective under acidic conditions and lasts approximately 30 min after being ingested. The sour modifying effects of this fruit were assayed in 13 subjects, where MB was shown to increase the sweetness of a low-calorie dessert without promoting subsequent energy compensation [40]. In addition, in another study, MB ameliorated insulin resistance in a preclinical model of diabetes induced by a chow diet enriched in fructose in rats. In a study where diabetes was induced using streptozocin (STZ), the glucose-lowering effects of miracle fruit administration was higher than that achieved by metformin treatment [41].

### 2.1. Antioxidant Activities of Miracle Berry Extracts

Most of the studies used to determine the composition of bioactive products from MB have focused on quantifying the total amount of polyphenols, including phenolic acids, flavonoids, and tannins, using the berry’s flesh as the starting raw material after extraction with polar solvents. Additionally, extracts from the seeds have also been evaluated, but berry flesh extracts contain higher amounts of phenols (up to five times) and flavonoids (up to three times) as compared with seed extracts.

Several studies have shown a positive correlation between the antioxidant activity, total phenolic content, and flavonoids.

In a study conducted by Inglett et al., 50% ethanolic extracts from skin, pulp, and seeds were compared with regard to their content of polyphenols and antioxidant activity. In this study, it was found that the higher antioxidant activities from skin- and pulp-derived extracts as compared with seed extracts were associated with the total content of polyphenols [35].

In another study, the antioxidant activity of MB flesh and seed-based methanolic extracts were evaluated via traditional free radical scavenging methods, i.e., ABTS (2,20-azino-bis(3-ethyl-benzothiazoline-6-sulfonic acid), DPPH (1,1-diphenyl-2-picrylhydrazyl), and FRAP (ferric reducing antioxidant power). Moreover, a fish oil emulsion model was used to measure the antioxidant activity of the extracts for avoiding lipid oxidation. In this study, the antioxidant activity of MB extracts was much higher than that of other well-recognized antioxidant-rich berries, such as blueberries and blackberries. Although in the ABTS and DPPH assays, the free radical scavenging activity of the MB flesh extract was similar to that of other antioxidant standards, in the FRAP assay, the activity of the flesh extract was significantly higher. Furthermore, the MB extract exhibited a greater ability to prevent lipid oxidation in the fish oil emulsion than gallic acid [42].

Extracts in the presence of 95% ethanol from the skin and pulp also showed positive correlations between the total phenolic content and the in vitro antioxidant activities [43].

In addition, methanol and chloroform extracts obtained from the complete fruit and pulp, respectively, showed anti-tyrosinase and antioxidant effects [44].

Few studies have described the identification of specific polyphenols from MB extracts. Methanolic extracts from the berry flesh [42] and leaves [45] have been partially characterized, demonstrating the presence of epicatechin, rutin, quercetin, myricetin, kaempferol, gallic, ferulic, syringic acid, anthocyanins (delphinidin glucoside, cyanidin galactoside, and malvidin galactoside), tocopherols (a-tocotrienol and a- and c-tocopherol), and lutein.

Water extracts from freeze-dried leaves have been shown to contain high amounts of bioactive phenolic components, including asp-hydroxybenzoic acid, vanillic acid, syringic acid, trans-p-coumaric acid, and veratric acid [46].

In this way, MB, by means of its antioxidant activity, may protect normal cells from carcinogenic transformations by reducing oxidative stress.

### 2.2. Miracle Berry Targeting Metabolic Risk Factors Associated with Cancer

For many years, plants have been considered to be a primary source of potent antidiabetic compounds. Many phytochemicals, such as flavonoids, carotenoids, alkaloids, saponins, phenolic acids, terpenoids, and glycosides have been demonstrated to exert antidiabetic effects through several mechanisms including insulin secretion, the inhibition of carbohydrate metabolizing enzymes, and the regeneration of pancreatic beta cells [47].

The effect of MB in ameliorating insulin resistance was first reported by Chen at al. [41] in a preclinical model of prediabetes in rats. Methanolic extracts obtained from the leaves also demonstrated antidiabetic effects including insulin synthesis, a reduction in inflammation, and the modulation of carbohydrate-metabolizing enzymes [48]. Ethanolic extracts from the pulp were shown to increase the uptake of glucose through the upregulation of the GLUT4 transporter in C2C12 myocytes [43], reinforcing the idea that MB could control the circulating levels of glucose. More studies are required to identify the specific bioactive compounds responsible for these effects and to demonstrate the safety of these compounds in terms of hypoglycemia.

In addition, ethanolic extracts from the seeds have shown potent anti-hypercholesterolemic activity, with lupeol acetate and β-amyrin-acetate triterpenoids identified as the main bioactive compounds responsible for this effect [49].

In another study, MB extracts, including water, butanol, ethyl acetate (EA), and hexane fractions were shown to reduce uric acid and inhibit xanthine oxidase activity in vitro. In this study, butanol extracts reduced oxonic acid potassium salt-induced hyperuricemia in ICR mice by lowering serum uric acid levels and activating hepatic xanthine oxidase [50].

Few studies have been conducted to evaluate the antitumoral activities in cancer cell lines. Extracts from the stems and the pulp were shown to reduce the proliferation of colorectal cancer cells, targeting c-fos and c-jun pathways [51]. In another study, extracts from the stems inhibited cell proliferation of melanoma cell lines [36].

Table 1 summarizes main studies conducted with extracts derived from miracle berry.

In summary, MB has been demonstrated to exert positive effects for controlling of metabolic risk factors in cancer, including anti-hypercholesterolemia and antidiabetic effects, together with antiproliferative effects in cancer cell lines.

## 3. Miraculin

The taste-modifying properties of MB are attributed to the presence of miraculin [37,41,52]. Miraculin sensitizes sweet taste receptors when sour acids are consumed [53], although miraculin itself does not taste sweet.

The clinical benefits of the taste-modifying effects of miracle fruits have been studied in the following two fields: (i) in cancer patients to ameliorate dysgeusia after chemotherapy and (ii) in low-calorie diets to reduce caloric intake in obese individuals [40,54,55].

Miraculin is a homodimeric glycosylated protein composed of 191 amino acid residues [56] with a maximum value of sweetness 400,000 times greater than that of sucrose on a molar basis [57]. The effect begins a few seconds after the fruit has been ingested and lasts between thirty minutes and two hours, after which miraculin dissociates from the taste receptors via the action of salivary amylase [58].

Miraculin interacts with class C sweet receptors, which consist of the heterodimers TAS1R2/TAS1R3. In the presence of acidic foods, these receptors undergo a conformational change allowing the glycans associated with miraculin to interact with the active centers of the receptors, and therefore activate them [59,60]. In this way, miraculin may act as a selective agonist (at acidic pH) or antagonist (at neutral pH) of TAS1R2/TAS1R3 sweet-taste receptors, depending on the pH value of the food or drink consumed [61].

Several intervention studies have shown that the degree of the taste-modifying effect differs according to the types and sources of the fruit or the type of foods to which MB is applied [62]. Miraculin, unlike other taste-modifying agents, is not sweet by itself but can change the perception of sourness to sweetness for a long period after consumption [63].

Importantly, miraculin along with thaumatin [64], curculin [65], brazein [66], and mabinlin [67] can be used as natural low calorie sweeteners with the potential to replace artificial sweeteners. Miraculin was first isolated by Kurihara and Beidler et al. from MB [53] and Brouwer et al. named it “miraculin” [68]. Due to the difficulty of growing MB in certain environments, genetic engineering has allowed the production of recombinant miraculin from other plants such as lettuce and tomato [69,70].

### 3.1. Mechanism of Action of Miraculin

Sweet taste receptors include G-coupled protein C receptors (GPCR) consisting of heterodimers of TAS1R2/TAS1R3 with the following three main domains: the transmembrane domain (TMD), the N-terminal segment Venus flytrap domain (VFD), and the cysteine-rich domain (CRD). Sucrose and glucose bind to the VFD of both the TAS1R2 and TAS1R3 subunits, whereas other sweeteners, such as aspartame and stevioside, interact only with the VFD of the TAS1R2 subunit but activate the G-coupled protein at the intracellular region of the TMD of TAS1R3 [71,72,73].

The proposed mechanism of action of miraculin starts with its association with the VFD of TAS1R2 as an inactive form at a neutral pH. Extracellular acidification induces partial activation of the sweet taste receptors through the interaction between extracellularly protonated TAS1R2 and the active form of miraculin. Complete activation requires weak acidification of the intracellular region [74]. The following intracellular cascade is common for the detection of other tastes such as umami and bitter. After TASR activation, the Gβγ subunit dissociates from the Gα subunit (α-gustducin) of the heterotrimeric G protein and activates phospholipase C-β2 (PLC-β2), which produces IP3. The interaction of IP3 with its receptor (type III IP3 receptor) activates TRPMP5 (transient receptor potential cation channel subfamily M member 5), releasing Ca^2+^ from the endoplasmic reticulum (ER). Then, the ATP channel Pannexin-1 (PX1) opens, releasing ATP to ultimately activate the gustatory afferent nerves [40,75,76,77] (Figure 2).

### 3.2. Clinical Applications of Miraculin

The harmful effects related to high sugar consumption are a matter of great public and scientific interest due to their association with the development of obesity and other chronic diseases, such as T2DM cardiovascular diseases (CVD) and cancer [78,79,80]. As a result, many alternative sweeteners have been investigated. However, there is still controversy regarding the harmful effects of consuming some artificial sweeteners, which may promote the development of glucose intolerance through the induction of compositional and functional alterations to the intestinal microbiota [81,82]. One health solution is the replacement of high-caloric sweeteners with alternative natural sweeteners that do not stimulate the intracellular insulin response [83]. Therefore, there is great interest in the use of natural substances as alternative sweeteners to prevent increases in body weight, fat mass, and blood pressure [84].

MB was reported to reduce the caloric intake in obese individuals on low calorie diets [85]. Miraculin enhanced the sweetness intensity of a low sugar dessert, thereby helping to limit the energy intake as compared with a high-sucrose supplemented dessert [86]. Another study evaluated the temporal profile of miracle fruit and its sugar substitute power in sour beverages through time intensity and temporal dominance of sensations tests. Unsweetened lemonade and lemonades with sugar, sucralose, and previous miracle fruit ingestions were also evaluated. Previous ingestion of miracle fruit provided high sweetness intensity and persistence and a sensory profile similar to that of sucralose [63]. In this way, the taste modifying effect of miraculin has great potential as an alternative sweetener or a taste modifier to mask undesirable sour tastes in food products.

Changes in taste perception are especially important in diseases such as cancer, where chemotherapy may alter taste perception (dysgeusia) [87]. It has been reported that taste alterations might be an early sign of tumor cell invasion in cancer patients [88]. Dysgeusia can lead to a loss of appetite and even malnutrition, which can effect patients’ quality of life and may also compromise the functionality of the immune system and the efficacy of the clinical treatments [89]. Moreover, as chemotherapy suppresses the immune system, which may augment the risk of infections, it is crucial to avoid malnutrition to preserve an effective immune system [90]. Two pilot studies (N = 8 and N = 23) were conducted to evaluate whether MB consumption before meals could ameliorate dysgeusia in cancer patients during chemotherapy treatment. Although no significant changes were found in the weights of the participants, MB improved taste perception, increasing food intake for some patients [54]. On the basis of the findings of the mentioned studies, it can be inferred that the use of MB in one’s diet may increase food’s palatability and reduce caloric intake at the same time, which makes it more difficult to suggest using this fruit or its derivatives in the diets of chemotherapy patients [55].

### 3.3. Miraculin Modulating the Gut–Brain Axis

In addition to the oral cavity, sweet taste receptors have been found in other organs, including the gastrointestinal tract (GI), the pancreas, adipose tissue, skeletal muscle, the bladder, and the brain [91].

Several studies have investigated the role of TASRs in the regulation of physiological functions at the GI level, including the regulation of GI motility and the secretion of enterohormones (leptin, ghrelin, insulin, GLP-1, and endocannabinoids), which regulate energy balance, systemic glucose levels, and food intake [92,93]. Enterohormones regulate satiety by acting locally in the GI tract or centrally through the gut–brain axis.

These effects have been proposed as useful therapeutic targets for the management of obesity, paving the way towards the pharmacological modulation of taste receptors as a promising and novel therapeutic strategy for the control of food intake in the management of obesity and T2DM [94].

In this sense, miraculin appears promising, by acting as a natural non-caloric sweetener and also as a TAS1R agonist at the GI level to modulate the release of distinct enterohormones [59].

TAS1R family members have been identified in distinct enteroendocrine cells, indicating biological and functional effects at the gastrointestinal level [95]. One such study compared insulin levels after the administration of glucose orally or intravenously. It was found that the levels of insulin were higher when glucose was administered orally than intravenously, revealing an effect of insulin secretion mediated by the effects of glucose at the GI level. In this regard, K and K cells are known to secret enterohormones such as GLP-1 and GIP, which mediate local and systemic functions [96]. Conversely, TAS1R2-TAS1R3 inhibition with lactisole reduced the levels of the secretion of the aforementioned incretins. Obese individuals have been shown to have reduced levels of the sweet taste receptor TAS1R3 [97], which may be due to the reduced glucose sensing in these individuals. In the same line, T2DM individuals experience diminished expression levels of sweet taste receptors TAS1R2 and TAS1R3, as well as their downstream regulator, α-gustducin [98].

Overall, manipulation of the responses mediated by intestinal sweet taste receptors is a promising field for developing therapeutic approaches against obesity or T2DM.

### 3.4. Nutrigenetics: Genetic Variability in the Perception of Sweet-Taste Affects Behavior and Nutrient Intake

The discovery of sweet taste receptors at the GI level and the ability of miraculin to interact with these receptors may have important consequences in the management of obesity or intestinal dysbiosis. Moreover, distinct polymorphisms in taste receptor genes have been demonstrated to influence food preferences and the risk of developing diet-related diseases, such as obesity [99,100]. In this way, sweet taste sensitivity has been recently related to BMI [101,102]. In a study conducted on obese individuals, reduced sweet taste perceptions was associated with active preferences for very sweet and fatty foods [103].

Several studies have investigated the associations between single nucleotide polymorphisms (SNPs) on sweet taste receptors, as well as several outcomes, including taste sensitivity, sugar intake, and interactions with the nutritional status of an individual, such as BMI. Dias et al. identified a genetic variant in the *TAS1R2* sweet taste receptor gene that was associated with sucrose taste sensitivity and sugar intake. Importantly, a significant gene–BMI interaction was found for both suprathreshold taste sensitivity and sugar intake. Individuals with a BMI ≥ 25 and carriers of the G allele of the rs12033832 SNP in the *TAS1R2* gene were less sensitive to sweet taste stimuli, suggesting increased sugar consumption [104].

Another study explored the association of four SNPs in the sweet taste receptor genes TAS1R2 (rs12033832 and rs35874116) and TAS1R3 (rs307355 and rs35744813) with sweet taste sensitivity and food intake in volunteers with a normal weight. In this study, the *TAS1R2* TT allele at rs35874116 was associated with a lower consumption of sweet foods as compared with C carriers. Conversely, *TAS1R2* AA carriers at rs12033832 consumed less energy from carbohydrates than the CC carriers, a result that challenged previous reports. Notably, the participants in the current study were all normal weight, as opposed to those in the work by Dias et al., which may explain the disparate results.

In this scenario, studying the interactions between miraculin and sweet taste receptors, together with analyses of the associations among several SNPs in sweet taste receptors and sucrose flavors, sugar intake, and metabolic alterations would help provide complementary tools for the management of metabolic disorders, including obesity, insulin resistance, and cancer.

## 4. Conclusions

The effects of MB at the systemic level (antioxidant, antidiabetic, anti-hypertriglyceridemic, anti-hypercholesterolemic, and anti-hyperuricemia) can help to reduce the metabolic stress associated with chronic diseases, such as obesity, metabolic syndrome, diabetes, and cancer. In a relevant way, the ability of miraculin to transform a sour taste into a sweet taste could be used as part of natural sugar reduction strategies. Preliminary studies also suggest that the positive effects of miraculin against dysgeusia can improve the perception of taste after chemotherapeutic treatments. Finally, due to the relevance of the stimulation of taste receptors at the GI level, specifically designed functional foods incorporating MB extracts (or isolated miraculin) represent an innovative potential strategy for the control of food intake to help prevent obesity and insulin resistance. Additional challenges involve investigating the following: (i) new extracts using sustainable technologies to expand the panel of bioactive compounds obtained from MB, such as supercritical fluid extraction using CO_2_; (ii) the interactions between SNPs in taste receptors, taste sensitivity, and associations with metabolic alterations; and (iii) for cancer patients, ameliorating dysgeusia and studying the possible synergism with conventional chemotherapeutic treatments in the clinic.

Figure 3 summarizes the main biological activities of miracle berry that could help alleviate metabolic stress in chronic diseases.

The use of miracle berry has been recognized by Japan’s Ministry of Health and Welfare. However, miraculin has not yet been legally recognized as a food additive by the U.S. Food and Drug Administration (FDA) or the European Union. The U.S. Department of Agriculture (USDA) does not place any restrictions on growing, selling, or eating miracle berry, and MB is currently being evaluated as a novel food in the EU by European Food Safety.

## Figures and Tables

**Figure 1 antioxidants-09-01282-f001:**
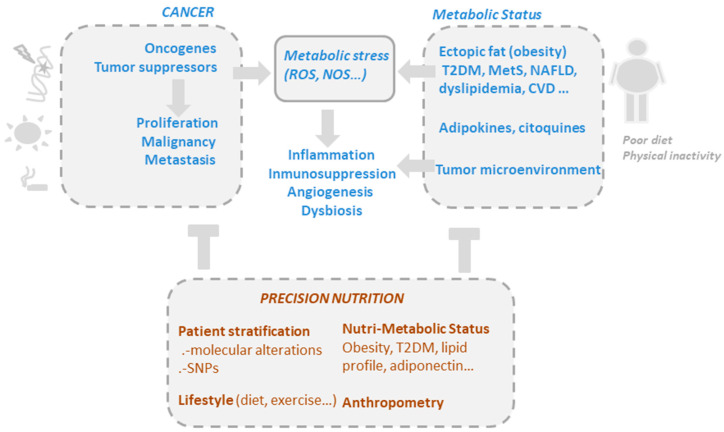
The drivers of metabolic and oxidative stress in cancer, including mutations in oncogenic pathways and risk factors such as obesity, poor diet, alcohol consumption, smoking, and a sedentary lifestyle. Precision nutrition aims to shape effective nutritional interventions based on scientific knowledge of the mechanism of action of bioactive compounds, taking into consideration the genetic variability and the metabolic and nutritional characteristics of individuals (nutri-metabolic scores), as well as lifestyle related factors.

**Figure 2 antioxidants-09-01282-f002:**
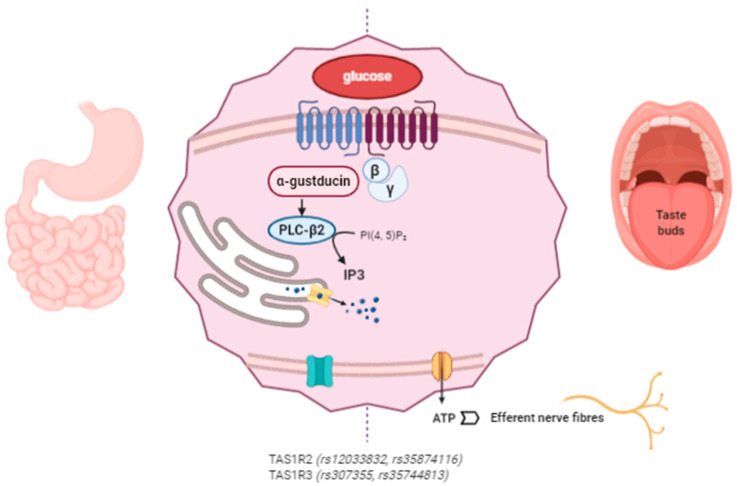
Miraculin interacts with sweet taste receptors TAS1R2 and TAS1R3 in the oral cavity and at the gastrointestinal (GI) level. The intracellular signaling pathways stimulate the taste receptors downstream under acidic conditions. Miraculin may contribute to the regulation of GI motility and the secretion of enterohormones (leptin, ghrelin, insulin, GLP-1, and endocannabinoids). The described SNPs in TAS1R affect taste sensitivity, sugar intake, and their interactions with body mass index (BMI). (The figure has been created by BioRender.com).

**Figure 3 antioxidants-09-01282-f003:**
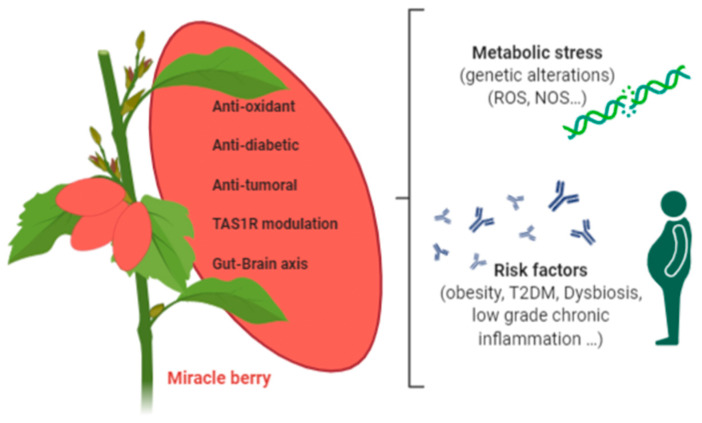
Summary of the main biological activities of miracle berry that could help alleviate metabolic stress in chronic diseases. (The figure has been created by BioRender.com).

**Table 1 antioxidants-09-01282-t001:** The main studies conducted with extracts derived from miracle berry.

Start Material	Solvent Extract	Effects	Type of Study	Ref.
Powder seeds	95% Ethanol	Cholesterol-lowering agent	Preclinical	[49]
Air-dried leaves	80% Methanol	Antidiabetic	Preclinical	[45,48]
Air-dried skin+pulp	95% EtOH	Uptake of glucose	In vitro (C2C12)	[43]
Miracle fruit powder	Water/butanol	Antioxidant; Anti-hyperuricemia	In vitro, Preclinical	[50]
Leaves, stems, and berries dried	80% MeOH, 10% EtOH	Reduce proliferation of colorectal cancer cell lines	In vitro	[51]
Stems	MeOH	Reduce proliferation in melanoma cell lines	In vitro	[36]
Pulp and fruit	MeOH-fruit	Anti-tyrosinase and antioxidation effects	In vitro	[44]
	CHCl_3_-pulp			
Miracle fruit powder	No extraction	Improve insulin resistance	Preclinical	[41]

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
