# Peer review of "Miracle Berry as a Potential Supplement in the Control of Metabolic Risk Factors in Cancer"

_antioxidants, 2020, doi:10.3390/antiox9121282_

Round 1
Reviewer 1 Report
This review, according to the title, aims to review the literature on Miracle Berry as co-adjuvant in the treatment of colo-rectal cancer.
There are some serious flaws in this review:
The title is misleading. The review is not about the use of MB as co-adjuvant in the treatment of colorectal cancer. I did not see any data on the use of MB as co-adjuvant (nor a comparison to existing strategies to treat colrectal cancer), nor any information on colorectal cancer.
The review lacks focus: it aims to adress cancer, but it is more on obesity and/or diabetes. Then, the focus shifts again to (disorders of) taste perception and an extensive discussion of miraculin (with no relation to colorectal cancer and/or diabetes).
There is absolute no information or discussion on how MB/miraculin would benefit 'precision nutrition'. The phrase 'precision nutrition' is completely misleading in this review.
The part on the gut microbiota is under-developed and obsolete.
A discussion on the value/importance of MB in any of the disorders/processes discussed in this review in comparison to other (natural) compounds is lacking. What is the added benefit of MB?
The review needs to be corrected by a native English speaker as there are many flaws in the use of English language
Keywords are missing
Author Response
Please see the attachmen

Reviewer 2 Report
Authors propose an overview on miracle berry as a potential co-adjuvant in colorectal cancer treatment in the frame of the Precision Nutrition.
The topic is actual and interesting, and the manuscript is quite well written and organized. However, some modifications are requested.
The manuscript requires an important editorial revision because instructions of the journal are not considered.
- Please modify abstract. It is too long and not completely fit with the organization of the paper
- Keywords are missing
- It could be better providing a list of abbreviations and use them after the first introduction in the text
- Capital letters in the title of the paragraphs according to instructions for authors
- Bibliographic references in the text inserted in square brackets
- Please modify references according to MDPI instructions
- Please check figure captions
Future directions cited in the abstract, such as the use of innovative extraction technologies, are not developed in the manuscript.
Paragraph 2.1 could be improved. In table 1 could be added information about type of extraction method used in addition to experimental conditions.
Reviewer 3 Report
For a review article, Figures and Tables are necessary because Figures and Tables could help readers to understand the topic easily. The manuscript only contains 2 figures and 1 table, authors could transfer many words as Figures and Tables. Furthermore, many chemical structures associated with Miracle Berry should be included in the manuscript. The review should be re-constructured.
Round 2
Reviewer 1 Report
I still have serious issues with this review.
Although the authors enhanced the focus of the review, it still is non-balanced: The authors appear to squeeze 3 topics in 1 review: 1) the relation between obesity and cancer; 2) the use of natural products to modulalte metabolism in cancer and 3) the action of miraculin on sweetness perception. In the revised version this part has a disproportional length.
As a result the review still lacks focus, balance and coherence. Each topic in itself would warrant a review.
There are still considerable problems with regard to the quality of the English text.
E.g. line 53: alarming should be alarmingly
Line 61, 64 and 91: the use of the word regardless: what is meant is actually the opposite: with regard to.
Line 99: manage should be management
And there are many more examples.
The entire mansucript should really be reviewed by a native English speaker
Author Response
Revised version R2 Antioxidants-1011898
Reviewer 1
I still have serious issues with this review.
- Although the authors enhanced the focus of the review, it still is non-balanced: The authors appear to squeeze 3 topics in 1 review: 1) the relation between obesity and cancer; 2) the use of natural products to modulalte metabolism in cancer and 3) the action of miraculin on sweetness perception. In the revised version this part has a disproportional length.
As a result the review still lacks focus, balance and coherence. Each topic in itself would warrant a review.
Thank you for the comment to increase the focus of the manuscript.
As the reviewer indicates, we need to restructure and to balance the three topics.
The idea of including the three topics is because we want to summarize the benefits of Miracle Berry further than its antioxidant activity in cancer (topic 1), but considering additional benefits with regard to the described biological effects of MB in ameliorating the metabolic stress associated to obesity and insulin resistance, which are known modifiable risk factors in cancer (topic 2). The third part aims (topic 3) to highlight the role of Miracle Berry, and more specifically, miraculin in the interaction with sweet taste receptors, not only at the oral cavity, but at the gastrointestinal level. We think this is a very promising field as it has been proposed additional benefits of Miraculin by mean of the modulation of the secretion of entero-hormones implicated in the intestinal motility, the secretion of insulin, and in saciety by mean of the gut brain axis. This will also contribute to ameliorate metabolic alterations which are risk factors in cancer.
We also agree that modulation of taste receptor at the gastrointestinal level is too long, and for this reason, we have reduced this part.
With these objectives, we have restructured the Manuscript, specifically, the abstract to better focus the manuscript. We have also rewritten the different sections, and we have eliminated some paragraphs to focus the manuscript.
.-There are still considerable problems with regard to the quality of the English text.
We are truly sorry for all the grammatical errors and the quality of the English.
For this reason, Manuscript has been revised by the MDPI English editing service checking grammar, spelling, punctuation and some improvement of style.
In addition, we have corrected all the indicated errors.
E.g. line 53: alarming should be alarmingly.
Sorry for the error. This has been corrected
Line 61, 64 and 91: the use of the word regardless: what is meant is actually the opposite: with regard to.
Sorry for the errors. These have been corrected as indicated.
Line 99: manage should be management
And there are many more examples.
The entire mansucript should really be reviewed by a native English speaker.
This has been revised as indicated by the MDPI English editing service.

Reviewer 3 Report
The article could be published in the present form.
Author Response
We want to thank the reviewer for all the suggestions which have contribute to improve the manuscript.
Round 3
Reviewer 1 Report
N.A.